# Characterization of Histone Deacetylase Expression Within In Vitro and In Vivo Bladder Cancer Model Systems

**DOI:** 10.3390/ijms20102599

**Published:** 2019-05-27

**Authors:** Jenna M. Buckwalter, Wilson Chan, Lauren Shuman, Thomas Wildermuth, Justine Ellis-Mohl, Vonn Walter, Joshua I. Warrick, Xue-Ru Wu, Matt Kaag, Jay D. Raman, David J. DeGraff

**Affiliations:** 1Department of Pathology and Laboratory Medicine, Pennsylvania State University College of Medicine, Hershey, PA 17033, USA; lshuman@pennstatehealth.psu.edu (L.S.); twildermuth1@pennstatehealth.psu.edu (T.W.); justinemarie0112@outlook.com (J.E.-M.); jwarrick@pennstatehealth.psu.edu (J.I.W.); ddegraff@pennstatehealth.psu.edu (D.J.D.); 2Department of Surgery, Division of Urology, Pennsylvania State University College of Medicine, Hershey, PA 17033, USA; Wilson.chan@unlv.edu (W.C.); mkaag@pennstatehealth.psu.edu (M.K.); jraman@pennstatehealth.psu.edu (J.D.R.); 3Department of Public Health Sciences, Pennsylvania State University College of Medicine, Hershey, PA 17033, USA; vwalter1@pennstatehealth.psu.edu; 4Department of Biochemistry and Molecular Biology, Pennsylvania State University College of Medicine, Hershey, PA 17033, USA; 5Departments of Urology and Pathology, New York University, New York, NY 10010, USA; Xue-Ru.Wu@nyulangone.org

**Keywords:** bladder cancer, heterogeneity, histone deacetylases

## Abstract

Epigenetic aberrations are prominent in bladder cancer (BC) and contribute to disease pathogenesis. We characterized histone deacetylase (HDAC) expression, a family of deacetylation enzymes, in both in vitro and in vivo BC model systems and analyzed expression data from The Cancer Genome Atlas (TCGA). Quantitative real-time polymerase chain reaction (qRT-PCR) and western blotting analysis was used to determine the expression status of Class I and II HDACs in ten human BC cell lines, while qRT-PCR was used to determine HDAC expression in 24 human tumor specimens. The TCGA cohort consists of 408 muscle invasive BC (MIBC) clinical samples and analysis of this data set identified expression of *HDAC4* and -*9* as being associated with basal–squamous disease. These findings agree with qRT-PCR results identifying increased expression of *HDAC4*, -*7*, and -*9* in basal BC cell lines (*p* < 0.05; Kruskal–Wallis test) and in clinical specimens with invasive bladder cancer (not statistically significant). We also observed increased expression in Hdac4, -7, and -9 in commonly used BC mouse models. Here, we identify suitable preclinical model systems for the study of HDACs, and show increased expression of Class IIa HDACs, specifically *HDAC4* and *HDAC9*, in basal BC cell lines and in invasive clinical specimens. These results suggest this class of HDACs may be best suited for targeted inhibition in patients with basal BC.

## 1. Introduction

It is estimated that 80,470 cases of bladder cancer (BC) will be diagnosed in 2019 [1]. Clinical management of BC poses several challenges. Specifically, early stage disease is often recurrent and manifests occasionally with progression to invasive disease. Thus, extensive follow-up is required for these patients, making treatment extraordinarily expensive [2,3]. Pathologic diagnosis of muscle invasive (MI) BC usually results in use of neoadjuvant, platinum-based chemotherapy (NAC) followed by cystectomy (bladder removal) [3]. Conventional tumor staging shows advanced BC is especially lethal, as the five-year survival rate of patients diagnosed with stage T2 disease is 63%, while patients diagnosed with stage T3 and T4 cancers exhibit significantly lower five-year survival rates of 46% and 15%, respectively [4]. In addition, ~50% of MIBC patients experience disease recurrence, usually in the form of metastatic disease which is almost uniformly lethal [3]. As it is currently impossible to identify which BC patients are at greatest risk for disease recurrence, progression, and death, further research is required to identify suitable prognostic markers and novel therapeutic targets to better manage the full spectrum of disease.

BC is a morphologically and genomically heterogeneous malignancy and several groups have competed a molecular characterization of early stage and advanced BC [2,5,6,7,8]. These studies have resulted in the description of prognostically significant transcriptional subtypes. The vast majority of early stage non-muscle invasive BC and approximately half of MIBC are broadly classified as exhibiting a “luminal” transcriptional signature. Luminal BC exhibits high expression levels of luminal urothelial markers, including forkhead box A1 (*FOXA1*), which is required for urothelial differentiation [9]. Interestingly, expression of *FOXA1* and other luminal markers is decreased in tumors classified as “basal” as well as in areas of morphologic squamous differentiation (SqD), which is enriched in basal disease [10,11]. Basal BC is extremely aggressive and patients with this subtype exhibit poor clinical outcomes. While the mechanisms that drive evolution to basal BC with SqD are generally unknown, epigenetic regulatory mechanisms are suspected to be contributing factors. 

There are a multitude of genetic alterations identified in both NMBIC and MIBC. These include loss of function mutations and copy number loss associated with tumor suppressors, and activating mutations and copy number gains associated with putative oncogenes [12,13]. For instance, NMBIC is considered relatively genomically stable and has a lower mutational frequency compared to MIBC. The most common genetic alteration in BC tumors is a deletion in chromosome 9 that results in copy number loss of cyclin dependent kinase inhibitor 2A (*CDKN2A*) and Tuberous Sclerosis complex subunit 1 (*TSC1*). In addition, inactivating mutations in tumor suppressor genes such as lysine demethylase 6A (*KDM6A*), cAMP-response element binding protein (*CREBBP*), and excision repair cross-complementation 2 (*ERCC2*) among other genes are important in the establishment and growth of NMIBC tumors. MIBC has a high somatic mutation frequency resulting in loss of function mutations in key tumor suppressors, including tumor protein 53 (*TP53*), retinoblastoma protein 1 (*Rb1*), phosphatase and tensin homolog (*PTEN*), AT-rich interaction domain 1A (*ARID1A*), E1A binding protein P300 (*EP300*), Ataxia–Telangiesctasia mutated (*ATM*), and lysine methyltransferase 2D (*KMT2D*). It is possible that genetic alterations of these genes may affect or cooperate with epigenetic modifiers driving changes in gene expression in BC tumors.

Epigenetic modifications including DNA methylation and histone post-translational modifications are closely associated with BC initiation and development, as reviewed in Reference [14], but perhaps most importantly, these modifications are dynamic, targetable, and potentially reversible [15]. Gene expression is, in part, mediated by acetylation status of histone tails where hyperacetylation drives transcriptional activation and hypoacetylation results in transcriptional repression [16,17]. These epigenetic states are tightly regulated by histone deacetylases (HDACs), a family of enzymes that deacetylate histones and other cellular proteins, and histone acetyltransferases (HATs), a family of enzymes that acetylate histones and other cellular proteins [18]. Belonging to the arginase/deacetylase superfamily, HDACs are classified into three different classes based on their structure and homology to yeast enzymes [19]. 

Genomic studies focused on late stage BC indicate an important role for alterations in epigenetic regulatory pathways [20], however, studies that characterized HDAC expression in BC have yielded conflicting results. Previous reports utilizing clinical specimens suggest *HDAC1*, -*2*, and -*3* are overexpressed in in urothelial carcinoma relative to normal urothelium [21,22,23]. However, an unrelated study suggested normal and cancerous bladder tissues exhibited similar expression levels for HDAC2 [24]. Furthermore, one additional study identified down-regulation or “variable expression” of *HDAC4*, -*5*, and -*7* when comparing urothelial cell carcinoma (UCC) cell lines and normal uroepithelial cells (NUC) [25]. While there are several potential reasons for these conflicting reports, the high degree of tumor heterogeneity at both the molecular and morphologic levels in this disease is potentially an important contributor [26,27,28]. In addition, a recent review summarized the most recent findings in regards to the role of HDACs in BC tumorigenesis and analyzed HDAC inhibitors (HDACis) as well as their success in clinical trials [29].

Previously, we classified ten BC cell lines into molecular subtypes and identified cell lines which exhibit genetic alterations and gene expression patterns consistent with luminal and basal molecular subtypes of human disease [9]. We also identified BC cell lines that were neither luminal nor basal subtype and are referred to as “non-type”. Using our previous molecular subtyping of BC cell lines and histological characterization of clinical specimens as well as genetically engineered mouse models (GEMMs), we characterized HDAC expression in commonly used in vitro and in vivo models of BC. 

## 2. Results

### 2.1. Class I and Class IIa Histone Deacetylases (HDACs) Are Overexpressed in Non-Type and Basal Human Bladder Cancer (BC) Cell Lines at the mRNA Level

We focused on the arginase/deacetylase superfamily which is comprised of three HDAC classes (Figure 1). Alterations in HDAC activity are significant contributors to the repression of gene expression in human cancers (reviewed in Reference [30]). Recent studies suggest this is potentially especially true in BC. Additionally, genomic studies in BC suggest alterations in epigenetic regulatory pathways may contribute to molecular and morphological intratumoral heterogeneity [31,32]. Therefore, we endeavored to determine the expression of different members of the HDAC superfamily in commonly used models of BC. Leveraging our previous study which subtyped ten BC cell lines [11], we performed quantitative real-time PCR (q-RT-PCR) to determine HDAC expression in models of luminal (RT4, SW780, and UMUC1) and basal–squamous (SCaBER, 5637, HT1376, and HT1197), as well as a subset of lines which did not fit into either luminal or basal–squamous subtyping schema (UMUC3, T24, and TCCSUP; hereafter referred to as “non-type”). 

Following the aggregation of cell line data into molecular subtype-specific groupings, we observed higher expression of Class I HDACs in both non-type and basal–squamous groups (Figure 2a–d). Specifically, *HDAC1* expression was highest in the basal–squamous group (Figure 2a; *p* < 0.05, Kruskal-Wallis test). Furthermore, *HDAC2* and -*8* were highly expressed in non-type and basal–squamous groups (Figure 2b,d; *p* < 0.05; Kruskal–Wallis test). In addition, increased *HDAC3* expression was detected in human BC cell lines included in the non-type molecular subtype grouping (Figure 2c; *p* < 0.05; Kruskal–Wallis test). We observed consistent high expression of Class IIa HDACs (Figure 2e–h) in non-type and basal molecular subtype groupings of human BC cells. Differences in expression of Class IIa HDACs among the molecular subtypes were statistically significant (*p* < 0.05; Kruskal–Wallis test) due to high expression of these genes in cell lines with non-type and basal molecular subtypes. Despite the significant association between Class HDAC expression and molecular subtype grouping, Class IIb *HDAC6* expression was not significantly different among the molecular subtypes (Figure 2i, *p* > 0.05; Kruskal–Wallis test). Interestingly, *HDAC10* expression was statistically significantly different among molecular subtypes of BC cell lines and expressed at higher levels in cell lines with a luminal subtype (Figure 2j, *p* < 0.05; Kruskal–Wallis test). In summary, specific members of the Class I and Class IIa group of HDACs are overexpressed in human BC cell lines classified as basal–squamous and non-type.

### 2.2. Histone Deacetylase 9 (DAC9) Is Expressed Primarily in Bladder Cancer (BC) Cell Lines with Non-Type and Basal Molecular Subtypes at the Protein Level

We next determined the degree of HDAC expression in human BC cell lines at the protein level in the same ten human BC cell lines (Figure 3). Our initial observations indicate HDAC1, HDAC2, HDAC3, HDAC6, HDAC7, and HDAC8 are expressed ubiquitously and independently of molecular subtype. Additionally, after densitometry analysis, we did not observe distinct trends by HDAC Class I and IIb for HDAC proteins. On the other hand, HDAC10 was expressed primarily in human BC cell lines with a non-type molecular subtype (Figure 3c. HDAC4 was expressed at the lowest levels of all examined enzymes, with highest expression levels exhibited in UMUC3. However, we do observe faint bands for HDAC4 in most human BC cell lines with a basal and luminal molecular subtype (Figure 3b,d). Interestingly, HDAC9 was the only HDAC expressed in human BC cell lines with a non-type or basal molecular subtype (Figure 3c,d). We were unable to identify a suitable antibody for HDAC5. A limitation of our analysis of HDAC proteins in human BC cell lines was using whole cell lysates. It may be advantageous to re-analyze HDAC proteins in cytoplasmic and nuclear extracts isolated from human BC cell lines. 

Because genetic alterations can impact expression and therefore functionality, we utilized the cBioPortal (http://www.cbioportal.org/) to access publicly available data through the Cancer Cell Line Encyclopedia study, enabling us to assess genetic alterations previously identified in *HDAC1-10* in a panel of BC cell lines [33]. In cell lines RT4, SCaBER, T24, and TCCSUP there are no mutations in *HDAC1-10*. In SW780 cells, *HDAC2* contains an inframe mutation and in UMUC3 cells, *HDAC4* contains a missense mutation. Copy number alterations were also observed in BC cell lines. For instance, *HDAC5* is amplified in 5637 cells and in HT1376 cells *HDAC9* is amplified, while *HDAC10* contains a deep deletion. Perhaps the most interesting observation of the genetic alterations of HDACs in these ten BC cell lines is *HDAC5*, the only HDAC to be mutated in more than one cell line. These results indicate that genomic alterations associated with genes encoding various HDAC members play a potential role in regulating the expression and/or function of these enzymes.

### 2.3. Basal–Squamous Bladder Cancer (BC) Clinical Samples Are Enriched for Histone Deacetylase (HDAC)4 and HDAC9 Expression

We interrogated publicly available data from The Cancer Genome Atlas (TCGA) in an effort to determine if changes in expression of Class IIa HDACs are associated with expression of luminal or basal BC markers. We observed high expression of *HDAC4* and *HDAC9* in basal–squamous tumors and clustered with expression of basal BC markers cytokeratins 14 (*KRT14*) and 5 (*KRT5*) (Figure 4a). Additionally, *HDAC4* and *HDAC9* expression is inversely correlated with Forkhead Box A1 (*FOXA1*), an important transcription factor in maintaining urothelial differentiation [9], and peroxisome proliferator-activated receptor gamma (*PPARG*), a steroid hormone receptor known for its oncogenic role in the development of BC [33]. 

Decreased expression and loss of both *FOXA1* and *PPARG* gene expression is observed in patient samples with basal BC enriched with SqD [34]. Therefore, it is possible that *HDAC4* and *HDAC9* may be a component of the regulatory complex that drives changes in expression of these important urothelial transcription factors in the development and progression of basal BC and development of SqD. Further analysis of the TCGA data set indicates statistically different expression of *HDAC4*, -*7*, and -*9* across molecular subtypes of human clinical specimens classified as neuronal, basal-squamous, luminal infiltrated, luminal, and luminal papillary (Figure 4b–d; *p* < 0.001, Kruskal–Wallis test). Luminal-infiltrated BC tumors express strong stromal and inflammatory signatures. Meanwhile, luminal tumors (without a modifier) have a stromal signature, but a minimal inflammatory signature. Luminal–papillary tumors lack both stromal and inflammatory signatures. Interestingly, higher expression of *HDAC4* and -*9* occurs in basal–squamous and luminal infiltrated molecular subtypes (Figure 4b,d). Further, while *HDAC7* expression is significantly different among molecular subtypes, this seems to primarily occur as decreased expression in neuronal subtypes and as increased expression in luminal papillary subtypes (Figure 4c). In conclusion, increased *HDAC4* and -*9* expression occurs in patients with basal–squamous disease which agrees with our observations in human BC cell lines with a basal molecular subtype. 

### 2.4. Bladder Cancer (BC) Driver Genes Correlate with Expression of Histone Deacetylases (HDACs)

As is the case with most cancers, genetic alterations in tumor suppressor genes and oncogenes contributes to BC tumorigenesis. Additionally, altered expression of urothelial differentiation factors such as *FOXA1* and loss of function mutations in chromatin remodeler proteins such as *ARID1A* are important in disease progression and contribute to tumor heterogeneity in BC. Therefore, we interrogated the TCGA bladder cancer study of 408 MIBC clinical samples to identify correlations between BC driver genes and HDACs (Table A1 and Table A2). 

The fact that activating mutations in phosphatidylinositol-4,5-bisphosphate 3-kinase catalytic subunit alpha (*PIK3CA*) and fibroblast growth factor receptor 3 (*FGFR3*) are common in NMIBC, as is copy number loss of *CDKN2A* suggest an important role for these genes in the development of early stage disease. Interestingly, *HDAC10* expression is positively correlated with the expression of *FGFR3* (0.43 Spearman correlation; *q* < 0.00001), while *HDAC10* expression is inversely correlated with the expression of *PIK3CA* and *CDKN2A* (−0.39 and −0.40 Spearman correlation respectively; *q* < 0.00001). Expression of *FGFR3* was also negatively correlated with *HDAC5* and *-9* (−0.25 and −0.40 Spearman correlation respectively; *q* < 0.00001). Interestingly, *PIK3CA* expression positively correlated with *HDAC4* and -*9* expression (0.27 and 0.23 Spearman correlation respectively; *q* < 0.00001), however, *PIK3CA* expression negatively correlated with *HDAC7* expression (−0.28 Spearman correlation; *q* < 0.00001). These results suggest that expression of genes commonly mutated in NMIBC correlate with distinct patterns of HDAC gene expression. Additionally, while increased or decreased expression of HDACs correlate with expression of specific driver genes in BC, these changes in expression do not cluster by HDAC class. 

As we previously mentioned, *FOXA1* is overexpressed in a luminal molecular subtype and loss of *FOXA1* expression is observed in a basal molecular subtype and areas of SqD. *FOXA1* expression positively correlates with *HDAC1*, -*6*, and -*10* (0.40, 0.26, and 0.35 Spearman correlation; *q* < 0.00001). Interestingly, *FOXA1* expression is inversely correlated with the expression of *HDAC4, -5*, and *-9* (−0.30, −0.27, and −0.37 respectively; *q* < 0.00001). These observations are interesting because *HDAC4, -5*, and *-9* are members of class IIa HDACs which we observed as being overexpressed in BC cell lines with a basal or non-type molecular subtype (Figure 2 and Figure 3) as well as in basal–squamous BC clinical samples (Figure 4). Decreased *FOXA1* expression also correlated with a basal molecular subtype in BC cell lines and in patients with basal–squamous disease. Thus, these observations suggest class IIa HDACs may play a role in regulating *FOXA1* expression in basal BC.

### 2.5. Histone Deacetylases (DACs) Exhibit Tumor Stage and Morphology-Specific Expression Patterns in Human Bladder Cancer (BC) Samples

Class IIa HDACs can be regulated via posttranslational modifications, the most extensively studied modification being phosphorylation, and by conformational changes that mask or expose nuclear localization or nuclear export signals [18,35]. Both mechanisms have a direct effect on HDAC functionality. Most importantly, Class IIa HDACs negatively influence the transcriptional process at the chromatin level and thus, need to be localized in the nucleus to associate with their respective macromolecular complexes [36,37]. Due to the interesting and complex functionality of Class IIa HDACs and their localization, we chose to focus on these HDACs for the remainder of this study. 

We categorized BC tumor tissue samples relative to tumor stage and morphology in a separate BC cohort and used q-RT-PCR to examine expression of *HDAC4*, -*7*, and -*9*. First, we characterized tumor tissue samples from radical cystectomies as non-invasive papillary BC (Non-Inv. UCC), muscle-invasive urothelial carcinoma (Inv. UCC), and muscle-invasive squamous cell carcinoma (Inv. SCC). While a subgroup of patients with Inv. SCC exhibited the highest expression of *HDAC4*, the highest mean levels of *HDAC4*, -*7* and -*9* expression were detected in NMIBC (Figure 5). When considering invasive disease alone, we observed the highest mean levels of *HDAC4*, -*7* and -*9* expression in Inv. SCC relative to Inv. UCC (Figure 5). Although these observations failed to reach statistical significance, they clearly warrant further investigation of Class IIa HDAC expression in tumor specimens that have been classified by both the molecular subtype and morphological characteristics.

### 2.6. Class IIa Histone Deacetylase (HDAC) Expression and Localization Are Altered in Commonly Used Bladder Cancer (BC) Mouse Models

We next sought to characterize Hdac4, -7, and -9 expression in some of the most commonly used mouse model systems for BC. In wild-type (WT) mice (Figure 6a), staining for Hdac4 and -7 was absent (Figure 6b,c), while positive cytoplasmic staining for Hdac9 was detected (Figure 6d). Exposure to environmental and chemical carcinogens, as well as carcinogens from tobacco smoke, represent potentially the best understood risk factor for the development of BC [38]. First identified as a bladder carcinogen in rodents [39,40], *N*-Butyl-*N*-(4-hydroxbutyl)nitrosamine (BBN) exposure in mice results in invasive tumors, and genetic alterations detected in human BC [41]. Therefore, we sought to characterize the effect of 20 weeks BBN (0.05%) exposure on Hdac4, -7, and -9 expression in the urothelium. Twenty weeks of *N*-Butyl-*N*-(4-hydroxbutyl)nitrosamine (BBN; 0.05%) exposure results in invasive tumors and SqD in adult mice (Figure 6e). While not detected in nuclei, we observed positive but weak cytoplasmic staining for Hdac4 and -7 in the areas of this tumor enriched with SqD (Figure 6f,g). Interestingly, we observed increased staining for Hdac9 in the cytoplasm and nuclei in this tumor compared to normal urothelium (Figure 6h). These observations suggest sequestering Hdac4 and -7 in the cytoplasm and increased nuclear Hdac9 expression may play a role in the development of tumors in mice treated with BBN. 

We also characterized Class IIa Hdac expression in several genetically engineered mouse models (GEMMs) of BC, including the *UpkII-Hras^*Q61L/WT^*, *UpkII-Simian virus 40* large T antigen (*SV40T*), and *UpkII-*SV40T/ *Hras^*Q61L/WT^* [42,43]. Consistent with previous reports [42], we observed urothelial hyperplasia (Figure 6i) in tissue obtained from an *UpkII-Hras^*Q61L/WT^* mouse model (hereafter simply *UPII-HRAS^*/WT^*), which harbors one copy of mutant *Hras*. While staining for Hdac4, -7, and -9 were all positive in the urothelium of *UpkII-HRAS^*/WT^* mice, Hdac4 staining was detected primarily in the cytoplasm (Figure 6j). Compared to WT urothelium, we observed increased Hdac7 and -9 expression in the cytoplasm and nucleus in all cell types of the urothelium and noticed some areas of the basal cell populations were particularly high (Figure 6k,l). These observations indicate Hdac7 and -9 may be preferentially upregulated in basal urothelium of hyperplastic tissue. As previously reported [43] we observed carcinoma in situ (CIS) in *UpkII-SV40T* mice (Figure 6m). While we observed increased Hdac4 expression in *UpkII-SV40T* mouse tumor tissue compared to WT urothelium, Hdac4 was primarily detected in the cytoplasm throughout the urothelium (Figure 6n). Comparatively, we observed increased expression of Hdac7 compared to WT urothelium, however, Hdac7 was also localized in the nucleus (Figure 6o). We also observed a dramatic increase Hdac9 expression compared to WT urothelium localized in the cytoplasm and nucleus with some higher areas of expression in the basal cell population (Figure 6p). These observations suggest Hdac4 and -7 localization in the cytoplasm and increased Hdac9 expression may contribute to development of CIS in this mouse model. 

*UpkII-SV40T/HRAS^*/WT^* mice developed hyperplasia and CIS (Figure 6q). Interestingly, we observed increased Hdac4 expression compared to WT mice localized in nuclei and cytoplasm (Figure 6r). Overall, we observed increased expression that was primarily cytoplasmic for Hdac7 and -9 (Figure 6s,t). Altogether, these results indicate combined expression of *SV40T* and oncogenic *Hras* in mouse urothelium alters Hdac4 and -9 expression and localization which may contribute to the development of CIS in this model.

We additionally examined Hdac expression in the previously described *UBC-Cre/ERT2/Foxa1^loxp/loxp^* mouse model [34]. This model results in inducible ablation of *Foxa1* in every layer of the urothelium and subsequently a loss of urothelial differentiation (hyperplasia (Figure 6u) and SqD). We observed increased cytoplasmic Hdac4 expression in *UBC-Cre/ERT2/Foxa1^loxp/loxp^* mice compared to WT mice (Figure 6v). Compared to WT urothelium, we also observed increased expression of Hdac7 in nuclei and cytoplasm of *Foxa1* ablated urothelium (Figure 6w), as well as increased expression of Hdac7 in basal and intermediate cell populations in these mice. Likewise, increased expression of Hdac9 in *UBC-Cre/ERT2/Foxa1^loxp/loxp^* mice was detected compared to WT mice (Figure 6x). Interestingly, Hdac9 was localized in the nucleus and cytoplasm, however, it was highly expressed in the basal cell population. These data indicate *Foxa1* ablation in mouse urothelium results in increased expression of Hdac4, -7, and -9. Moreover, Hdac7 and -9 localize to the nucleus suggesting these factors may play a role in gene expression changes observed with *Foxa1* loss. 

## 3. Discussion

Here, we identify suitable preclinical model systems for the study of HDACs in BC. Specifically, Class I HDACs consist of *HDAC1, -2*, *-3*, and *-8*. Overexpression of Class I HDACs has been observed in various cancers including those of the colon, prostate, and bladder [23,44,45,46]. With the exception of HDAC8, Class I HDACs are localized primarily in the cell nucleus. Class II HDACs are capable of shuttling between the nucleus and cytoplasm. Within Class II there are two subclasses: Class IIa (*HDAC4*, *-5*, *-7*, and *-9*) and Class IIb (*HDAC6*, *-10*). Class IV consists only of *HDAC11* which was not studied in this work. Based on other findings and for the purpose of simplicity, we did not investigate the deoxyhypusine synthase-like NAD/FAD-binding domain superfamily, which consists of Class III HDACs.

We found Class IIa HDACs were more likely to be overexpressed in non-type and basal molecular subtypes in vitro and in clinical samples with squamous cell carcinoma (SCC). Specifically, we show increased expression of Class IIa HDACs, particularly *HDAC4* and *-9*, in basal BC cell lines and in invasive SCC human clinical specimens. Additionally, we observed increased expression of Class IIa HDACs in commonly employed BC mouse models as well as altered cellular localization of HDACs. We also observed increased expression and altered cellular localization of Hdac4, -7, and -9 in commonly used BC mouse models. Our results suggest these specific members of the Class IIa HDACs could be potentially targeted for disease management. 

We characterized the expression of HDACs in commonly used BC cell lines which identified expression of *HDAC4*, -*7*, and -*9* at the mRNA level (Figure 2e,g,h) and HDAC9 at the protein level (Figure 3a,c,d) as having increased expression in non-type and basal molecular subtype. Our observations were confirmed when analysis of the TCGA bladder study identified expression of *HDAC4* and -*9* as correlated with expression of basal markers (Figure 4a) and patients with basal-squamous and luminal infiltrated BC (Figure 4b,d). 

Increased expression of *HDAC*4 and -*9* in patients with invasive SCC further supported the idea that Class IIa HDACs are important in the development of a basal molecular subtype and SqD in BC. While less than 5% of BC cases are purely SCC, components of invasive UCC are often enriched with areas of SqD [10,11]. Thus, HDAC4, -7, and -9 may be a component of an epigenetic regulatory complex that alters the gene expression profile of a luminal cell that results in a basal molecular subtype and may even contribute to the development of SqD. Alternatively, HDAC4, -7, and -9 may simply correlate with SqD. Interestingly, in mouse models known to develop SqD or high-grade disease including BBN treated mice (Figure 6e–h), *UpkII-SV40T* (Figure 6m–p), *UpkII-SV40T/HRAS^*/WT^* (Figure 6q–t), and UBC-Cre/ERT2/Foxa1^loxp/loxp^ (Figure 6u–x) we observed overexpression of Hdac4 and/or Hdac9 as well altered cellular localization suggesting in these models Hdac4 and -9 are important in the development of hyperplasia and/or CIS in vivo. 

The clinical benefit of HDAC inhibitors (HDACi) on patient outcome has been limited and some clinical trials resulted in severe toxicities in patients [47]. However, the targeting of HDACs seems to be a promising therapeutic approach for BC, as in vitro and in vivo studies, as well as some clinical trials, have had some success [48,49]. Perhaps the best opportunity for the use of HDACi clinically is to be used as one component of a multi-drug regimen. In regards to availability of HDACi, in addition to HDACi that specifically target Class I HDACs, there is a set of HDACi that target *HDAC6* [50]. There are also several pan-HDACi under development. Further supporting the likely use of HDACi as one component of a multidrug approach, treatment of BC cell lines with HDACi in isolation results in growth inhibition, cell cycle arrest, and reduced proliferation [44,51], while combining HDACi with other additional therapeutic approaches resulted in apoptosis and cell death [52]. While these studies highlight the benefit of combining other already established drug therapies with HDACi to maximize anti-tumor, studies such as our current one investigating HDAC expression in commonly used in vitro and in vivo preclinical models is important. This is true because in order to investigate the anti-tumor capability and direct effects of a specific HDACi, the HDAC expression profile of the BC model system needs to be well defined. 

We specifically used the TCGA bladder study to determine the correlation of expression of *HDAC4*, *-7*, and -*9* with molecular subtypes as well as histological and pathological subtypes in BC (Figure 4). A recent review on HDACs in BC also employed the TCGA bladder study to examine HDAC mutations and expression [29]. Here the authors identified HDAC somatic mutation frequency to be low (0.2–2.4%). In addition, the authors observed multiple HDAC family member mutations and alterations within a single solid tumor. Altogether, these observations combined with the well-established concept of tumor heterogeneity in BC allows for understanding of how specific gene expression patterns or acquired mutations of HDACs contribute to disease pathogenesis. However, our observations of *HDAC4* and -*9* expression in basal molecular subtypes in vitro and in clinical specimens are important in defining the specific roles HDACs may play in establishment and progression of disease. 

In summary, we have characterized the most commonly used BC preclinical model systems in regard to HDACs expression. Our results demonstrate that increased expression of Class IIa HDACs, in particular *HDAC4* and -*9*, occurs in basal BC cell lines and invasive clinical specimens suggesting this class of HDACs may be best suited for targeted inhibition in patients with basal BC. Studies such as these are important and necessary to develop further understanding of HDAC mechanistic contributions to BC intratumoral heterogeneity that negatively impacts therapeutic efficacy. 

## 4. Materials and Methods

### 4.1. Cell Culture

All bladder cancer cell lines were purchased from the American Type Culture Collection (ATCC), except UMUC1 bladder cancer cells (Sigma Aldrich, St. Louis, MO, USA). All cell lines are routinely screened for mycoplasma infection (PromoKine, Heidelberg), and authenticated via STR analysis (Genetica, Burlington, NC, USA). The bladder cancer cell lines RT4 and T24 were cultured in McCoy’s Modified Medium (Corning, Tewskbury, MA, USA) with 10% Fetal Bovine Serum (FBS) (Atlanta Biologicals, Flowery Branch, GA, USA). UMUC1 and UMUC3 bladder cancer cells were cultured in Minimal Essential Medium (GE Healthcare Life Sciences, Pittsburgh, PA, USA) supplemented with 10% FBS. SCaBER, HT1197, HT1376 and TCCSUP bladder cancer cell lines were cultured in MEM following the addition of Non-Essential Amino Acids and 10% FBS. The bladder cancer cell lines 5637 and SW780 were cultured in Roswell Park Memorial Institute 1640 (RPMI 1640; Corning) following the addition of 10% FBS. 

### 4.2. Western Blotting

All cell lysates were prepared using RIPA lysis and extraction buffer (Thermo Fisher Scientific). Protein concentrations following cell lysis were measured by using the PierceBCA Protein Assay Kit (Thermo Fisher Scientific, Waltham, MA, USA) as per manufacturer’s instructions. Following extraction, protein samples consisted of 13 µg of protein, 1X loading dye sample buffer (Thermo Fisher Scientific), and 10% 2-mercaptoethanol (Sigma) were electrophoresed on 4-12% Bis-Tris NuPAGE gels (Thermo Fisher Scientific), and proteins were subsequently transferred to nitrocellulose blotting Membrane (GE Healthcare Life Sciences) using a Pierce G2 Fast Blotter (Thermo Fisher Scientific) according to manufacturer’s instructions. Following transfer, membranes were incubated at room temperature in 5% non-fat milk (NFDM) dissolved in Tris buffered saline containing 0.1% Tween-20 (TBST; Bio-Rad, Hercules CA) for 1 hour. Additionally, all primary antibodies used in this study were diluted in TBST with 5% NFDM. Dilutions of primary antibodies were as follows: anti-HDAC1 (1:1000; D5C6U, Cell Signaling Technologies, Danvers, MA, USA), anti-HDAC2 (1:1000; D6S5P, Cell Signaling Technologies), anti-HDAC3 (1:1000; D2O1K, Cell Signaling Technologies), anti-HDAC4 (1:1000; D8T3Q, Cell Signaling Technologies), anti-HDAC5 (1:1000; D1J7V, Cell Signaling Technologies), anti-HDAC6 (1:1000; D2E5, Cell Signaling Technologies), anti-HDAC7 (1:1000; D4E1L, Cell Signaling Technologies), anti-HDAC8 (1:1000; ab187139, Abcam, Cambridge, MA, USA), anti-HDAC9 (1:1000; ab59718, Abcam), anti-HDAC10 (1:1000; ab108934, Abcam), and anti-GAPDH (1:2000; 14C10; Cell Signaling Technologies). After incubation with primary antibodies overnight at 4°C degrees all membranes were washed 5 times for 5 minutes with TBST. Secondary antibody (ECL anti-rabbit, HRP-linked whole antibody; 1:2000; GE Healthcare Life Sciences) was diluted in TBST containing 5% NFDM and incubated at room temperature for 1 hour. After incubation with secondary antibodies, membranes were washed 5 times for 5 minutes with TBST. Protein bands were visualized by exposing membrane after addition of enhanced chemiluminescent Western Blotting Substrate (Pierce) to X-ray film (Thermo Fisher Scientific) via standard procedures.

### 4.3. RNA Extraction and Quantitative Real Time Polymerase Chain Reaction (Q-RT-PCR)

All human clinical specimens were used following approval of the Pennsylvania State University College of Medicine Institutional Review Board. Formalin fixed paraffin embedded (FFPE) tumor tissue samples were obtained from radical cystectomies of 9 patients with non-invasive papillary UC, 15 patients with muscle-invasive UC (pT2-4, N0/1), and 12 patients with muscle-invasive squamous carcinoma (pT2-4, N0/1). RNA extraction from cell lines and human clinical specimen was performed using RNeasy (Qiagen, Valencia CA) as per manufacturer protocol. Q-RT-PCR was performed using QuantaStudio 7 Real-Time PCR System (Applied Biosystems) using a 96-well format. Using 500 ng of RNA, cDNA was prepared using the M-MLV reverse transcriptase kit (Thermo Fisher Scientific) according to the manufacturer’s protocol. Reactions consisted of 5 µL of cDNA per reaction, 10 µL of 2× Taqman Gene Expression Master Mix (Applied Biosystems, Foster City CA) and 1 µL of 20× Taqman probe, as well as nuclease-free water (total reaction volume 20 µL/well). The following Taqman probes with corresponding catalogue number for human genes were used in this study. HDAC1 (Hs00606262_g1), HDAC2 (H200231032_m1), HDAC3 (Hs00187320_m1), HDAC4 (H201041648_m1), HDAC5 (H200608351_m1), HDAC6 (H200997427_m1), HDAC7 (Hs01045864_m1), HDAC8 (Hs00954353_g1), HDAC9 (Hs01081558_m1), and HDAC10 (Hs00368899_m1). Relative gene expression was analyzed using deltadeltaCt method [53] using 18 S ribosomal RNA as a reference.

### 4.4. Statistical Analysis

The Kruskal–Wallis test was applied to compare HDAC expression levels in BC cell lines and patient tissue is based on groups defined by molecular subtype [9] and conventional histomorphometric examination, respectively. RNAseq-based gene expression data for the TCGA BLCA cohort (*n* = 408) was downloaded from the Broad Firehose GDAC (https://gdac.broadinstitute.org/). Gene expression subtypes, histological subtypes, and squamous pathology classifications for the TCGA BLCA cohort were obtained from the Supplementary Data of Robertson et al. (2018) [32]. The Kruskal–Wallis test was applied to compare HDAC expression levels in bladder cancer cell lines, bladder cancer patients, and the TCGA cohort based on groups defined by the molecular categorization of [9], conventional histomorphometric examination, and gene expression subtype, respectively. R 3.5.0 (R Core Team) was used to generate figures and perform all statistical tests [54].

### 4.5. Immunohistochemistry

Immunohistochemistry (IHC) was completed as previously described [34]. Briefly, slides were deparaffinized and rehydrated using a series of graded alcohols followed by washing with running tap water for 5 minutes. Antigen retrieval was performed by placing slides in 1% antigen unmasking solution (Vector Labs, Burlingame, CA) and heating the slides at high pressure in a pressure cooker for 20 min (Cuisinart CPC-600FR). To prevent boiling and preserve tissue integrity, steam was released in short bursts. The slides were cooled to room temperature and washed 3 times for 10 minutes in phosphate-buffer saline (PBS, pH 7.4). Unless otherwise noted, all incubations were performed at room temperature. Blocking of endogenous peroxidases was achieved by incubating slides in 1% hydrogen peroxide in methanol for 20 minutes. The slides were then washed 3 times for 10 minutes in PBS. Sections were incubated in PBS containing horse serum (Vector Labs) for 1 hour in an effort to reduce nonspecific antibody binding. Slides were incubated overnight with primary antibody at 4 °C in a humidified chamber. Primary antibodies used for IHC include mouse anti HDAC4 (1:200; Santa Cruz Biotechnology, Santa Cruz, CA, USA), rabbit anti HDAC7 (1:100; Cell Signaling Technologies), and rabbit anti HDAC9 (1:200; Abcam). After overnight incubation with primary antibody, slides were washed 3 times for 10 minutes in PBS. Tissue sections were incubated in biotinylated secondary antibody diluted in PBS containing horse serum (1:200; Vector Labs) for 1 hour. Binding specificity of each antibody was visualized using Vectastain Elite ABC Peroxidase Kit (Vector Labs) according to the manufacturer protocol with diaminobenzidine substrate buffer as the chromogen (Dako). 

## Figures and Tables

**Figure 1 ijms-20-02599-f001:**
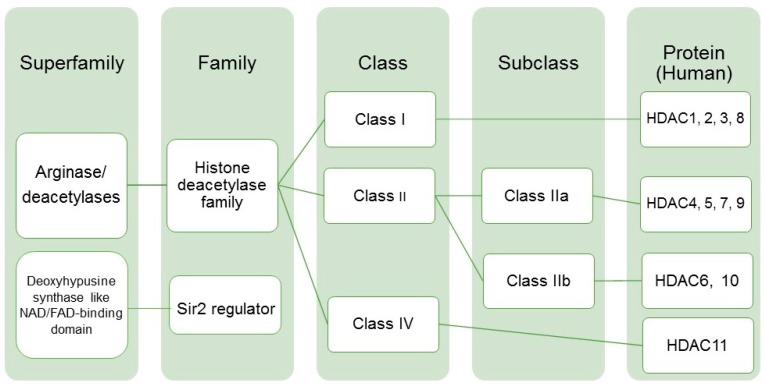
Histone deacetylase family/classes of proteins and expression in human bladder cancer tumors. Histone deacetylase family and respective classes and subclasses with human specific proteins (reviewed in [19]).

**Figure 2 ijms-20-02599-f002:**
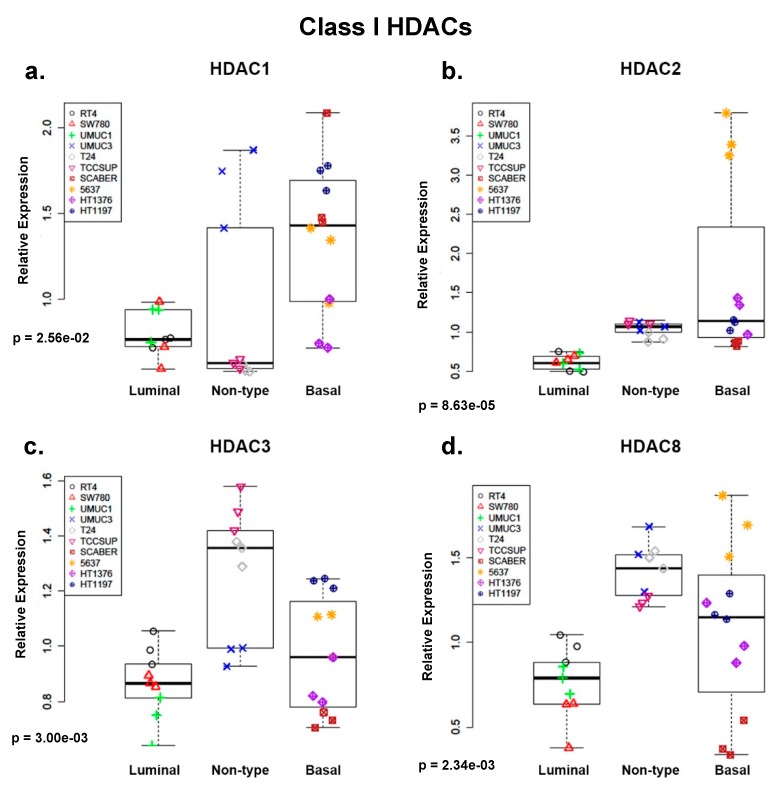
*HDAC2*, -*3*, -*4*, -*7*, and -*9* are overexpressed in basal-squamous and non-type human BC cell lines. Boxplots depicting histone deacetylase (HDAC) expression in 10 bladder cancer (BC) cell lines (three replicates per cell line, with individual replicates depicted as one point). The cell lines are grouped based on the molecular characterization as previously described [9]. (**a**–**d**) Overall, class I HDAC expression increases in non-type and basal subtypes of human BC cell lines with significant differences in *HDAC2* and -*8* expression in human BC cell lines with non-type and basal subtypes compared to human BC cell lines with a luminal subtype. *HDAC1* overexpression was observed only in human BC cell lines with a basal subtype compared to luminal and non-type molecular subtypes. *HDAC3* overexpression is observed primarily in human BC cell lines with a non-type subtype. (**e**–**h**) Class IIa HDAC expression was consistently increased in human BC cell lines with non-type and basal molecular subtypes compared to luminal human BC cell lines. (**i**,**j**) Class IIb HDAC expression varied independent of molecular subtype in human BC cell lines.

**Figure 3 ijms-20-02599-f003:**
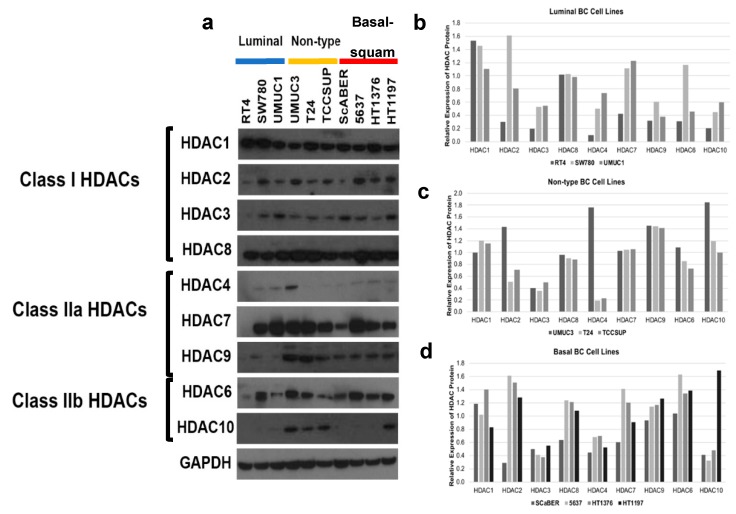
*HDAC9* is overexpressed in basal-squamous and non-type human BC cell lines. (**a**) Western blotting of histone deacetylase (HDAC) expression in a panel of 10 human BC cell lines, classified as luminal (RT4, SW780, UMUC1), non-type (UMUC3, T24, TCCSUP) and basal–squamous (SCaBER, 5637, HT1376, HT1197). Densitometry is depicted in (**b**–**d**). Expression of HDAC1, 2, 3, 6, 7, and 8 appeared to be independent of molecular subtype assignment in human BC cell lines. HDAC4 protein was detected at low levels in SW780, UMUC1, UMUC3 and in 5637, HT1376, and HT1197. On the other hand, HDAC9 was robustly expressed in non-type and basal cell lines, with low levels detected in luminal lines. Interestingly, HDAC10 was expressed primarily in human BC cells with a non-type molecular subtype.

**Figure 4 ijms-20-02599-f004:**
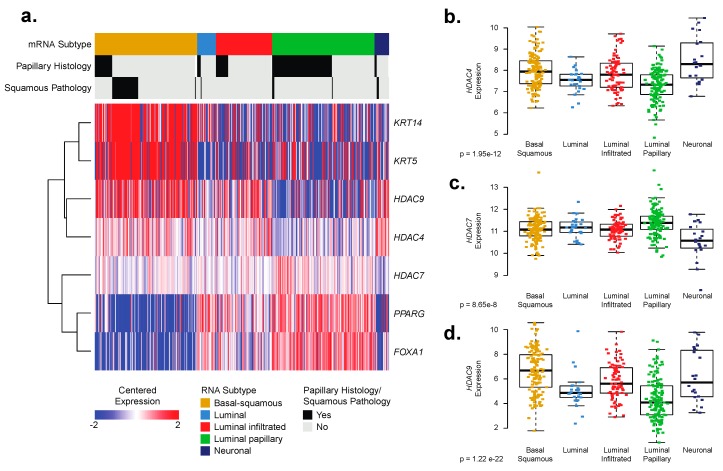
Histone deacetylase (HDAC4) and HDAC9 expression co-clusters with basal markers in muscle-invasive bladder cancer (BC). (**a**) Heat map display of expression values (log2(normalized RSEM + 1)) from the Cancer Genome Atlas (TCGA) bladder cancer cohort (*n* = 408) for *HDAC4*, *HDAC7*, and *HDAC9*, as well as select genes that serve as markers of luminal (forkhead box A1 (*FOXA1*), peroxisome proliferator-activated receptor gamma (*PPARG*)) and basal (*KRT14*, *KRT5*,) BC. Hierarchical clustering was applied after median centering the expression values by gene. Annotation tracks show gene expression subtype, histological subtype, and squamous pathology classification. (**b**) *HDAC4* is overexpressed in neuronal, basal-squamous, and luminal infiltrated subtypes. (**c**) While statistically significant among subtypes, *HDAC7* occurs as decreased expression in neuronal subtypes and increased in luminal papillary subtypes. (**d**) *HDAC9* is overexpressed in basal–squamous and luminal infiltrated subtypes.

**Figure 5 ijms-20-02599-f005:**
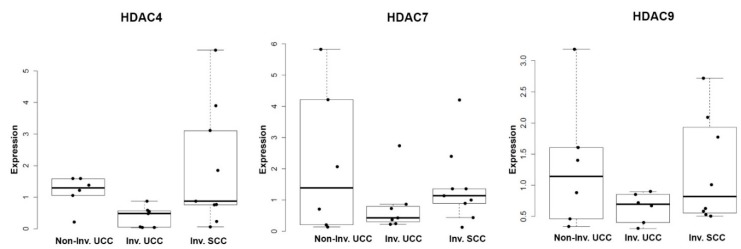
Quantitative real-time polymerase chain reaction (PCR) analysis of Class IIa histone deacetylase (HDAC) expression in bladder cancer (BC) clinical specimens. Boxplot displays of HDAC expression from 24 BC patients. Patients are grouped according to conventional histomorphometric examination (see materials and methods). However, while we note a trend of increased expression of *HDAC4*, -*7*, and -*9* in invasive squamous cell carcinoma (SCC) these differences are not statistically significant.

**Figure 6 ijms-20-02599-f006:**
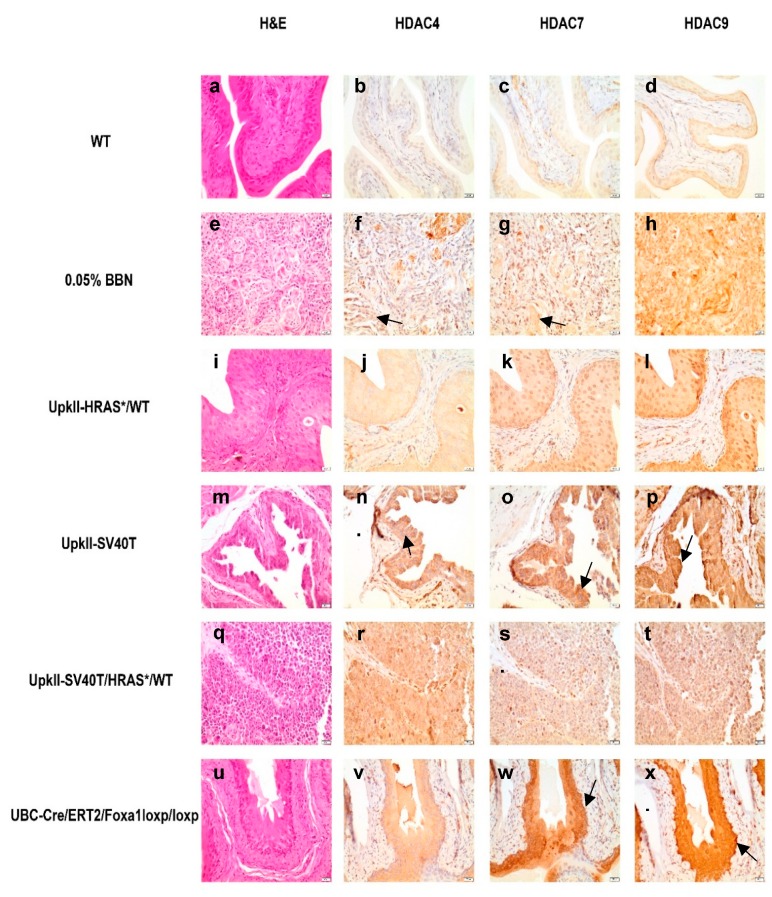
Increased expression of histone deacetylase (HDAC)4, -7, and -9 is observed in commonly used bladder cancer BC mouse models. In normal mouse epithelium (**a**), HDAC4 (**b**) and -7 (**c**) is not expressed and HDAC9 is positive (**d**). In mice treated with 0.05% *N*-Butyl-*N*-(4-hydroxbutyl)nitrosamine (BBN) for 20 weeks (**e**), we observed increased expression of HDAC4 (**f**) and -7 (**g**) in the cytoplasm, while Hdac9 was overexpressed in nuclei and cytoplasm (**h**). In *UpkII-HRAS^*/WT^* mice (**i**), we observed increased expression for all three HDACs (**j**–**l**). In *UpkII-SV40T* mice (**m**), Hdac4 (**n**) and -7 (**o**) were overexpressed in the cytoplasm and Hdac9 (**p**) was overexpressed in the nuclei and cytoplasm. In *UpkII-SV40T/HRAS^*/WT^* mice (**q**), we observed overexpression of Hdac4 (**r**) in nuclei and cytoplasm as well as increased expression in Hdac7 (**s**) and -9 (**t**) but primarily in the cytoplasm. Finally, in *UBC-Cre/ERT2/Foxa1^loxp/loxp^* mice (**u**), we observed increased Hdac4 expression (**v**), however, Hdac7 (**w**) and -9 (**x**) were overexpressed with intense staining observed in the basal cell populations. Black arrows are pointing towards area of interest as described.

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
