# Peer review of "Characterization of Histone Deacetylase Expression Within In Vitro and In Vivo Bladder Cancer Model Systems"

_ijms, 2019, doi:10.3390/ijms20102599_

Round 1
Reviewer 1 Report
This is a well written and informative manuscript. My concerns and comments are listed below;
1. Abstract: It would be helpful to state the number of TGCA patient datasets that were assessed as part of this study, the number of bladder cancer cell lines used, and p values for the associations reported in the abstract. As is, the abstract is too vague
2. Introduction: The introduction section should make note of other genetic alterations other than increased expression of FOXA1 that contribute to bladder cancer progression, and make note of if/how HDACs influence the expression of these. I recommend that expression levels of these molecules (e.g. ATM and p53; per a recent IJMS review, these molecules can be HDAC targets in bladder cancer cells (https://www.ncbi.nlm.nih.gov/pmc/articles/PMC6471041/, the authors should consider citing this paper))
3. Results/discussion section: The authors of the IJMS paper mentioned above also conducted TGCA analysis of bladder cancers/assessed HDAC expression; please compare and contrast the findings from this study and their study
4. The authors should consider performing statistical analyses which compare HDAC expression with the expression of genes known to drive bladder cancer to determine if a correlation exists. As a follow up, acetylation studies could be performed. As is, the study is very descriptive in nature – it simply documents the expression levels of various HDACs in different models.
5. Figure 6: please include arrows pointing to areas of interest that are noted in the figure legend
6. Results section: HDAC mutations occur frequently in bladder cancers. The authors should make note of the mutational status of the HDACs they are assessing in the various cell lines and GEMMs as this can impact function.
7. Results section: it would be informative to assess the localization of the HDACs in the cell lines. This data is included for the GEMM analysis; it appears there are differences in localization of the HDACs.
8. Discussion section: more time should be spent relating the information to other studies of HDACs/bladder cancer. Again, the article above should be mentioned/discussed. Lines 329 – 331 state ‘However, the targeting of HDACs seems to be a promising therapeutic approach for BC as in vitro, in vivo, and clinical trials have had some success [45].’ The authors should expand on this/discuss these data in light of the own findings.
Author Response
Comments are in uploaded word document.

Reviewer 2 Report
In this manuscript, Buckwalter et al. evaluated the differential expression profiles of histone deacetylases in various molecular subtypes of bladder cancer. By investigating the expression of these important epigenetic regulators on the transcriptomic and proteomic levels in bladder cancer cell lines, they reported a unique enrichment of HDAC9 in the squamous/basal subtype. Analysis of publicly available gene expression profile data from patients supported this finding and found HDAC4 and HDAC9 to be highly expressed in the basal subtype as opposed to the luminal. However, these findings were not significantly reproducible in a separate cohort consisting of patients suffering from invasive squamous cell carcinoma. Finally, the pattern of expression of HDAC4, 7, and 8 was evaluated by IHC in various bladder cancer mouse models. In general, Buckwalter et al. provide useful information concerning the diverse levels of expression of HDACs in different molecular subtypes of bladder cancer. Most importantly, they meticulously investigate these levels in numerous systems such as cell lines, mouse models, and patient data. However, this manuscript endeavors to underscore a specific pattern of enrichment of a certain class of HDACs in the squamous subtype which the data does not clearly support. Minor changes in the way the data is presented will ensure the clear interpretation of the findings of this manuscript which I think is worthy of publication after these modifications. These suggested alterations are in the following:
1- Significance is not shown for the qPCR data in Figure 2. While the significance is mentioned in text, it is necessary to show the significance on the figures as well to allow for accurate interpretation of the data. Also the Y-axis is not labeled clearly (only as expression) while labeling it as relative mRNA expression is more accurate and enables the reader to understand the figure without having to refer to the whole text.
2- The densitometry data shown in Figure 2b-d does not correlate with the western blot shown in Figure 2a. For example, RT4 is shown to have almost no expression of HDAC3 in the western blot but the bar indicating the densitometry is quite high and is very similar in height to the bar indicating expression in SW780 which is shown to have a higher expression and comparable GAPDH loading. While normalization to loading control is a good idea, it should not introduce so much bias that the cells lacking the expression are shown to have the same levels of other cells with clear expression. Also, the grouping of the data based on the cell line rather than the HDAC makes it seem like the comparison is possible in the same cell line regarding the expression of the different HDACs. This is not accurate as these regulators are detected using different antibodies and cannot be compared to each other. Grouping the data based on HDACs in different cell lines can enable the reader to compare the expression of the same HDAC in different lines which is what the authors aim to as they would like to detect enrichment of a certain HDAC in certain molecular subtype.
3- As the aim of Figure 4a is to show that HDAC4 and HDAC9 but not HDAC7 correlate with squamous pathology and markers, why not to sort the heatmap based on pathology or mRNA subtype? While the hierarchical clustering between the genes is very fitting, the hierarchical clustering between patients is unnecessary and decreases the tendency. The scale bar only shows high and low expression while the values considered to be high or low are not mentioned. Significance is also missing from the boxplots in Figure b-d. Moreover, luminal papillary and infiltrated are not explained enough in the text. It would be of great assistance if the authors can add few sentences about these subtypes and their significance.
4- The authors state that they observed an increased expression of HDAC4 and -9 in invasive SCC samples while Figure 5 clearly shows that the expression of HDAC4 and -9 is less than that of non-invasive UCC while all the checked HDACs show a slightly higher expression compared to invasive UCC. The authors may consider toning down the title and also changing the way they are presenting or explaining this result. For example, it looks like there is a subgroup of patients who indeed express higher levels of HDAC4. It is naturally expected that not all patients will show the same pattern and it will still be very interesting to study these factors if only a subgroup of patients with the squamous molecular subtype highly express it.
5- The authors may like to add arrows indicating the tumor in Figure 6 to make it easier for the reader to know where the tumor is. In line 244, the authors state that “These observations suggest sequestering Hdac4 and -7 in the cytoplasm and increased nuclear Hdac9 expression may contribute to the development of SqD in mice treated with BBN”. This sentence tends to be more of an overstatement as the development of the tumor was not studied and the levels of HDACs were not evaluated after development.
Author Response
Comments are in uploaded word document.
